# Genetic Factors in Antiphospholipid Syndrome: Preliminary Experience with Whole Exome Sequencing

**DOI:** 10.3390/ijms21249551

**Published:** 2020-12-15

**Authors:** Alice Barinotti, Massimo Radin, Irene Cecchi, Silvia Grazietta Foddai, Elena Rubini, Dario Roccatello, Savino Sciascia, Elisa Menegatti

**Affiliations:** 1Center of Research of Immunopathology and Rare Diseases—Coordinating Center of Piemonte and Aosta Valley Network for Rare Diseases, S. Giovanni Bosco Hospital, Department of Clinical and Biological Sciences, University of Turin, 10154 Turin, Italy; alice.barinotti@unito.it (A.B.); massimo.radin@unito.it (M.R.); irene.cecchi@unito.it (I.C.); silviagrazietta.foddai@unito.it (S.G.F.); elena.rubini@unito.it (E.R.); linotipico@gmail.com (D.R.); elisa.menegatti@unito.it (E.M.); 2Department of Clinical and Biological Sciences, School of Specialization of Clinical Pathology, University of Turin, 10125 Turin, Italy; 3Nephrology and Dialysis, Department of Clinical and Biological Sciences, S. Giovanni Bosco Hospital and University of Turin, 10154 Turin, Italy

**Keywords:** antiphospholipid syndrome, immunogenetics, whole exome sequencing, network-based approach, antiphospholipid antibodies, thrombosis, autoimmune diseases, autoantibodies

## Abstract

As in many autoimmune diseases, the pathogenesis of the antiphospholipid syndrome (APS) is the result of a complex interplay between predisposing genes and triggering environmental factors, leading to a loss of self-tolerance and immune-mediated tissue damage. While the first genetic studies in APS focused primarily on the *human leukocytes antigen system (HLA)* region, more recent data highlighted the role of other genes in APS susceptibility, including those involved in the immune response and in the hemostatic process. In order to join this intriguing debate, we analyzed the single-nucleotide polymorphisms (SNPs) derived from the whole exome sequencing (WES) of two siblings affected by APS and compared our findings with the available literature. We identified genes encoding proteins involved in the hemostatic process, the immune response, and the phospholipid metabolism (*PLA2G6*, *HSPG2*, *BCL3*, *ZFAT*, *ATP2B2*, *CRTC3*, and *ADCY3*) of potential interest when debating the pathogenesis of the syndrome. The study of the selected SNPs in a larger cohort of APS patients and the integration of WES results with the network-based approaches will help decipher the genetic risk factors involved in the diverse clinical features of APS.

## 1. Introduction

The antiphospholipid syndrome (APS) is an autoimmune disease defined by the occurrence of venous and/or arterial thromboses and/or recurrent pregnancy morbidity (such as miscarriages, fetal deaths, and premature birth), in the presence of the persistent positivity for antiphospholipid antibodies (aPL) [1]. These autoantibody specificities include lupus anticoagulant (LA), anti-cardiolipin antibodies (aCL), and anti-β2 glycoprotein-I antibodies (anti-β2GPI), even though new ones, such as anti-phosphatidylserine/prothrombin antibodies (aPS/PT) are emerging as an additional tool to be considered in APS diagnosis [2,3,4].

To date, our understanding of this rare disorder is still evolving, but it is common knowledge that it can occur as an isolated disorder, the so-called “primary APS” (PAPS), and that it can also be associated with other autoimmune diseases, such as systemic lupus erythematosus (SLE) and referred as “secondary APS” (SAPS). In less than 1% of patients, it can also develop into a catastrophic form (CAPS), which is a life-threatening condition leading to massive thrombotic complications that can affect multiple organs and systems [5].

The etiology of APS is still unknown, but similarly to other autoimmune diseases [6,7,8,9,10], it seems to be linked to a complex interplay between genetic predisposition, antigenic stimuli, and the presence of specific autoantibodies. Various proofs of the genetic predisposition of APS are available: from the familiar clustering of cases, to affected monozygotic twins, a greater prevalence of aPL in the serum of subjects sharing the same descent of patients, animal models, and association with various alleles such as those associated with the human leukocytes antigen system (HLA) [11,12,13,14,15,16,17,18]. With regard to this latter aspect, it seems that the genetic predisposition is in part due to the HLA system [11]. The contribution of immunogenetics to the development of aPL and APS has been addressed mainly by family studies and by population studies looking at the HLA region, but recent data highlighted the role of variations affecting *non-MHC genes* in APS susceptibility [11,19,20]. Among these, there are other genes involved in the immune response outside the HLA region and genes that take part in inflammatory and hemostatic processes [21,22].

To date, studies investigating the genetic bases of diseases have been revolutionized by the development of massive parallel sequencing techniques such as next-generation sequencing (NGS), which is a powerful tool that is able to test simultaneously, in a single sample, thousands of genomic variants. The whole-exome sequencing (WES) is a successful tool in genomic research: exome is the best-characterized part of the human genome (about 1%), it is the protein-coding part, and it is predicted to harbor about the 85% of the mutations that are known to have large effects on disease-related traits, especially in Mendelian disorders. While on one hand WES allows easily identifying mutations causative of known monogenic diseases, on the other hand, the identification of genome variants involved in known complex diseases that show an unusual familial clustering is more challenging and often an unexplored application. Family studies represent one of the possible approaches to genetic studies, focusing on genotypes seen in affected members and not present in unaffected ones, leading to the identification of new mutations involved in the development of a pathologic phenotype. This kind of application of WES could be particularly interesting with regard to pathologies whose precise genetic background is still unclear, such as the APS.

Herewith, we aim to discuss the state of the art and share our preliminary experience in investigating the genetic background of APS through NGS data derived from APS family studies.

## 2. Genetics of APS: What We Know

The familial incidence of aPL has been documented since the 1980s [23,24,25], and the first genetic studies about APS focused on the HLA region; thus, this region was the earliest genetic association described in this disorder, as well as in most autoimmune-mediated diseases.

Familial and population studies revealed that the loci that are most likely implicated in conferring susceptibility to the development of aPL and APS are *HLA-DR4*, *DR7*, *DR9*, *DR13*, *DR53*, *DQ6*, *DQ7*, and *DQ8*, and in particular, those most represented across several ethnic groups seem to be *HLA-DR4* and *HLA-DRw53* [26]. Despite these findings, the identification of specific HLA alleles associated with APS remains a challenge, largely because of the small sample sizes achieved from this rare disorder, which often leads to results with low statistical significance and the high complexity of the HLA region.

To date, several efforts have been made in order to better elucidate the genetic background of APS, and different studies have also demonstrated the possible role of variations affecting *non-MHC genes* in APS susceptibility. One of the first genetic risk factors for APS found outside the HLA region was a polymorphism in the β2GPI gene. A recent meta-analysis [27] revealed an association of β2GPI Val/Leu247 polymorphism and APS, and functional studies found a correlation between this variant and the production of anti-β2GPI antibodies [28], supporting the hypothesis that *B2GP1* could take part in APS development.

Similarly to what has been observed in SLE [29], other genes outside the HLA region, but involved in the immune response, seem to have a role in APS. *STAT4*, the gene encoding the signal transducer and activator of transcription-4, is one of the genetic factors thought to have a role in multiple autoimmune diseases, such as rheumatoid arthritis and SLE [30,31]. Even if its precise role in APS pathophysiology remains unclear, its polymorphisms have been found in APS patients and seem to correlate with an increased sensitivity to interferon-α (IFNα) [32,33]. The same has been observed for other genes involved in the immune response, such as *BLK proto-oncogene* (*BLK*) [34], *Interferon regulatory factor 5* (*IRF5*), and *Protein tyrosine phosphatase non-receptor type 22* (*PTPN22*), even if the role of these two latter genes in APS remains still controversial [35,36,37].

As previously reported, thrombotic events represent the main clinical manifestation of APS. Patients affected by APS show a pro-coagulant phenotype due to a dysregulated activation of platelets, endothelial cells, and monocytes in association with the disruption of physiological anticoagulant and fibrinolytic systems. Other genes that might play a role in the etiology of APS include genes involved in the inflammatory response, such as *Toll-like receptor 4* (*TLR4*) and *Toll-like receptor 2* (*TLR2*) [38,39] and in platelets adhesion, such as *Integrin subunit alpha 2* (*GP Ia*) and *Integrin subunit beta 3* (*GP IIIa*) [40,41] in patients that underwent thrombotic events. Other genes include those involved in the coagulation cascade, such as *protein C receptor* (*PROCR*) and *protein Z-dependent inhibitor* (*ZPI*) [42,43].

Moreover, in a recent study, Perez-Sanchez et al. [44] when comparing the genetic profile of patients affected by isolated APS, APS plus SLE, SLE and healthy controls, differences in the gene expression profile between the three diseases were found. In fact, in this study, 20–30% of the genes differentially expressed in the three conditions were related to atherosclerosis, inflammation, and cardiovascular diseases. Focusing on APS, we also observed, even if smaller, a difference between PAPS and SAPS patients. In particular, primary APS seem to be more associated with alterations in mitochondria biogenesis and function and oxidative stress, while secondary APS showed a bigger correlation with IFN expression and genes mediating atherosclerotic and inflammatory signaling.

Finally, of great interest is also a recent work by Islam et al. [45] that provided further insights into the investigation of genetic risk factors for the development of thrombotic manifestations in APS patients. The authors conducted a systematic research focusing on case-control studies assessing the risk of the association of a particular gene in the development of thrombosis in patients with PAPS. Starting from a total of 2673 articles, they finally included 22 studies, consisting of a total of 1268 PAPS patients and 1649 healthy controls. From these 22 studies, they found 16 genes associated with thrombotic PAPS: *PF4V1 (Platelet Factor 4 Variant 1*), *SELP (Selectin P*), *TLR2 (Toll-like receptor 2*), *TLR4 (Toll-like receptor 4*), *SERPINE1 (Serpin family E member 1*), *B2GP1 (Beta-2-glycoprotein I*), *GP Ia (Integrin subunit alpha 2*), *GP1BA (Glycoprotein Ib platelet subunit alpha*), *F2R(Coagulation factor II receptor*), *F2RL1 (Coagulation factor II receptor-like 1*), *F2(Coagulation factor II*), *TFPI (Tissue factor pathway inhibitor*), *F3 (Coagulation factor III*), *VEGFA (Vascular endothelial growth factor A*), *FLT1 (FMS-related tyrosine kinase 1*) and *TNF (Tumor necrosis factor*) [41,46,47,48,49,50,51]. Interestingly, a protein–protein interaction analysis showed that the resulting encoded proteins seem to be inter-connected and that they are mainly involved in two biological processes: the coagulation cascade and the immune response. Moreover, the 16 genes resulted to be expressed in 32 different organs, dealing with the fact that the thrombotic events can affect multiple organs and systems of the human organism. Table 1 summarizes the common mutations that have been associated with APS to date.

## 3. APS Family Study

### 3.1. Methods

#### 3.1.1. Patients Selection

Two family members, attending the San Giovanni Bosco Hospital (Turin), affected by APS, were identified. These two siblings, patient 1 (female; 51 y.o.) and patient 2 (male; 47 y.o.), were both triple positive for aPL. When analyzing their clinical history, patient 1 manifested a serious case of CAPS, while patient 2 presented with recurrent thrombotic venous events. More detailed information on the clinical features of the patients enrolled in the study are shown in Table 2.

Complete clinical history was collected from patient’s notes, and genetic-based research was performed on the two affected patients and their father (who had a negative clinical history for thrombosis and tested negative for aPL). No information was available from the mother, who died a few years ago from a neoplasia. A written consent to the publication of this study was given by the three patients.

#### 3.1.2. Exome Sequencing

Genomic DNA was extracted from peripheral blood using DNeasy Blood & Tissue Kit (Qiagen). The samples underwent exome capture by an Agilent V6 exome kit and were then sequenced by the DNB-SEQ PE100 platform. Sequencing was done on a 100× coverage, and reads have been aligned to the human reference genome (GRCh37/hg19 assembly) using the Burrows–Wheeler Alignment tool (BWA) [52]. The mean sequencing depth on target regions was 170× for patient 1, 205× for patient 2, and 221× for the father. Moreover, 99.50% of the targeted bases had at least 10× coverage for all the three donors. Variant calling process was performed by the Genome Analysis Toolkit (GATK) [53] and Variant Studio software (illumina) was used for variants annotation.

#### 3.1.3. Variants Filtering, Annotation, and Prioritization

First, single-nucleotide polymorphisms (SNPs) were analyzed. In order to select the most promising variants resulted from the WES, we applied a filter according to their population frequency using the Genome Aggregation Database (https://gnomad.broadinstitute.org/), with a 1% cut-off.

Second, a cross-match comparison between patient 1 and patient 2 SNPs was performed in order to obtain the SNPs shared by the two affected siblings. Therefore, the identified SNPs were compared with the father’s to obtain the variations shared by the two patients, but not in common with their father, as a healthy control (Figure 1).

Third, the SNPs that were not affecting protein function and that were scored as benign were removed. This was evaluated according to VarSome (https://varsome.com/) which, integrating the knowledge of more than 70 genomic databases, provides information regarding the predicted effect of DNA variants on the protein, defining variants as either pathogenic, likely pathogenic, of uncertain significance, likely benign, or benign [54].

Missense variants were checked for coverage by the Integrative Genomics Viewer software (IGV http://software.broadinstitute.org/software/igv/) [55]. This approach showed a coverage far greater than 20×, considering the minimum value for variant significance, for each SNP.

For prioritization and selection of the most promising variants, a complete literature research has been performed in order to obtain data on the candidate genes and mutations.

### 3.2. Results

#### 3.2.1. SNPs of Interest Found in Patient 1 and Patient 2

After the step-by-step screening process reported in the Methods section, the SNPs number was significantly reduced, comprehending 27 missense mutations, on which further analyses have been focused (Table 3).

The complete literature research regarding the genes carrying the missense mutations led to further reducing the list of selected genes, focusing on those that exert a role potentially involved in APS pathogenesis and development.

##### PLA2G6

The *Phospholipase A2 group VI* (*PLA2G6*) gene product is iPLA2β, which is a phospholipase Ca2+-independent that is involved in several biological pathways, such as phospholipids remodeling, arachidonic acid release, leukotriene and prostaglandin production, and apoptosis [56]. Of particular interest is its role in cardiolipin (CL) metabolism. Song et al. [57] demonstrated that iPLA2β failure led to an impaired remodeling, from which can arise several events that could be an in vivo trigger for aPL generation (mitochondrial membranes damage, cyt C release to the cytosol, oxidative stress, apoptosis, and CL and its metabolites’ exposure to membrane surface).

Given the known role of another family of phospholipase A2 [the cytosolic PLA2 (cPLA2)] in complement activation and C5b-9 membrane attack complex assembly on glomerular epithelial cells (GECs), some studies investigated if this function can also be attributed to the iPLA2 family. When focusing on iPLA2β, Cohen et al. showed that iPLA2β seems to attenuate the complement-induced GECs injury, which probably takes place through a sort of preconditioning action on these cells that makes them more resistant [58].

Analyzing more in detail the polymorphism present in patients 1 and 2, it is also worth noting that the position of this mutation seems to be relevant. The mutation is in a hydrophobic domain that is closely interconnected with the catalytic one, and it involves a hydrophobic amino acid that becomes polar non-charged [59].

##### HSPG2

*Heparan sulfate proteoglycan 2*(*HSPG2*) encodes for perlecan, which is a proteoglycan that represents the main component of the vascular extracellular matrix. It is synthesized by the vascular endothelial cells and it is known to be important in maintaining the endothelial barrier function and vascular homeostasis [60]. In particular, recent works showed its anti-proliferative effect on the smooth muscle cells, its anti-thrombotic effect after a vascular injury, and also its role in atheroprotection by the activation of endothelial nitric oxide synthase (eNOS) expression [61,62,63].

Moreover, it has been observed that after vascular injury, Caspase-3 activation of apoptotic endothelial cells leads to a proteolysis-mediated release of a truncated C-terminal fragment of perlecan that harbors a laminin G motif: the so-called LG3. This peptide can also be released by infiltrating leukocytes and aggregating platelets at the site of vascular damage and inflammation [64]. Recent data highlighted that the production of antibodies directed against LG3 (anti-LG3) can induce vascular remodeling. Studies focusing on renal transplantation showed that the presence of anti-LG3 increase the ischemia–reperfusion damage and thus the renal dysfunction, prompting the activation of the classical complement pathway that enhances microvascular injury [64,65].

##### BCL3

The product *BCL3 transcription coactivator* (*BCL3*) is known to be associated with B cell leukemia/lymphoma and it is a member of the IkB (inhibitors of kB proteins) family. Bcl3 shows a regulatory role in NF-kB pathways, which play a pivotal role in immune system development and the induction of successful immunity, since the majority of NF-kB target genes encode immunomodulatory proteins (e.g., cytokines and chemokines) and proteins involved in antigen presentation [66].

In particular, altered expression levels of Bcl3 has been observed in autoimmune diseases (such as rheumatoid arthritis, Crohn disease, and autoimmune diabetes), and it shows important functions at different levels of the immune response, from the organogenesis of secondary lymphoid organs (SLOs) to antigen presentation and antibody class-switching. Bcl3 deficiency seems to lead to defects in the microarchitecture of SLOs and to increase the susceptibility to infections, while Bcl3 overexpression can induce alterations of both B and T cells compartments [66,67]. Moreover, a study conducted by Weyrich et al. also showed Bcl3 involvement in fibrin expression and retraction [67].

##### ZFAT

*Zinc finger and AT-hook domain containing* (*ZFAT*) is a zinc finger protein; it is a transcriptional regulator involved in apoptosis and cell survival, and it resides in a susceptibility locus for autoimmune thyroid disease. Its protein is predominantly expressed in B and T cells, and it plays a role in the regulation of genes important in the immune response. In particular, Koyanagi et al. [68] observed that ZFAT overexpression in B cells progenitors resulted in the downregulation of a subset of other genes involved in the immune response. The authors also showed that ZFAT deficiency leads to an impaired positive selection in the thymus [68].

In addition to these functions related to the immune response, it seems also to be expressed in endothelial cells and to take part in vascular remodeling, since its deficiency has been associated with an impaired capillary-like network formation in human umbilical vein endothelial cells (HUVECs), with a worst impairment correlated to a greater deficiency [69,70].

##### ATP2B2

Plasma membrane Ca2+ATPase pump (PMCA2) is the protein encoded by *ATPase plasma membrane Ca2+ transporting 2* (*ATP2B2*) gene and it belongs to the family of P-type primary ion transport ATPases, which remove bivalent calcium ions from cells and play a critical role in intracellular calcium homeostasis. This pump presents 10 transmembrane domains, 2 large intracellular loops, and 2 cytoplasmatic tails (N-terminal and C-terminal) [71,72]. Its function can be influenced by the presence of calmodulin, whose binding removes the auto-inhibition, activating the pump and by the phospholipid composition of the surrounding plasma membrane. Acidic phospholipids enhance the sensitivity of the pump to calcium: the stimulatory potency is proportional to the number of negative charges.

PMCA2 seems to play a role in vascular homeostasis, regulating nitric oxid (NO) production in the endothelial cells: in particular, Holton et al. observed that eNOS activity was negatively regulated by PMCA2 [73].

##### CRTC3

*CREB-regulated transcription coactivator 3* (*CRTC3*) is a member of the CREB regulated transcription coactivator gene family. This family regulates cAMP response element-binding protein (CREB)-dependent gene transcription in a phosphorylation-independent manner, and it may be selective for cyclic adenosine monophosphate (cAMP)-responsive genes. A recent work [74] suggested that CREB activation can play role in vascular smooth muscle cells (VSMCs), showing both an anti-mitogenic and a mitogenic effect. In particular, Hudson et al. [74] demonstrated that CREB activation by serine-133 phosphorylation leads to the proliferation of VSMCs, while its activation by CRTC2 and CRTC3 in response to cAMP stimuli promotes a growth arrest. Moreover, CREB seems to be crucial in maintaining basal endothelial monolayer integrity and in restoring normal endothelial barrier function after an increase of permeability caused by inflammatory mediators such as thrombin, VEGF, histamine, and lipopolysaccharide (LPS); thus, the mutation of its coactivators could also influence these processes [75].

##### ADCY3

*Adenylate cyclase 3* (*ADCY3*) product is primarily implicated in obesity onset. The downregulation of ADCY3 is associated with metabolic dysfunctions such as increased visceral adiposity, reduced expression of genes involved in thermogenesis, fatty acids oxidation and insulin signaling, and increased plasma levels of pro-inflammatory cytokines [76,77].

#### 3.2.2. Analysis of Genes Previously Associated with Thrombosis

An analysis focusing on genes known to be associated with thrombophilia has also been conducted on patient 1 and patient 2. In particular, the following genes have been analyzed: *F5* (*Coagulation factor V*), *F2* (*Coagulation factor II*), *MTHFR* (*Methylenetetrahydrofolate reductase*), *F13A1* (*Coagulation factor XIII A chain*), *PROC* (*Protein C*), *PROS1* (*Protein S*), *FGB* (*Fibrinogen Beta chain*), and *SERPINE1* (*Serpin family E member 1*).

Neither patient 1 nor patient 2 had known mutations associated with a pro-thrombotic state affecting these genes [78,79,80,81,82], and also, no “new” mutations that could influence the genes function have been observed.

Other genes previously associated with thrombotic primary APS and recently reported by Islam et al. [45] have been analyzed (i.e., *PF4V1*, *SELP*, *TLR2*, *TLR4*, *GP Ia*, *GP1BA*, *F2R*, *F2RL1*, *TFPI*, *F3*, *VEGFA*, *FLT1*, and *TNF*). Patient 1 and patient 2 did not show mutations affecting the above genes.

#### 3.2.3. Analysis of B2GPI Gene

When analyzing the *β2GPI* gene, it has been observed that the two patients shared six SNPs on this gene, but all of them had a frequency much higher than 1%, which was “benign” according to VarSome, and none of them were previously associated with APS.

## 4. Discussion and Future Perspectives

APS is a complex rare disease with the great majority of the cases being sporadic. Rarely, the condition has been reported to run in families.

What can familiar cases of APS teach us?

To date, the majority of the genetic studies about APS were performed in small cohorts of patients. As a result, only a few genetic associations reported are convincing, such as those with *HLA class II* alleles and the clinical diagnosis of APS. Recent insights into the genetic components of APS are promising to contribute to our knowledge and understanding of the possible molecular pathways involved in the development of this disease.

Our preliminary experience with a family case showed a selection of genes of potential interest being related to the immune response and the vascular homeostasis, thus providing new attractive areas for further investigation. As most autoimmune diseases, the genetic background in APS is multifactorial, impacting on different levels of cellular components.

In this study, when analyzing the genetic profile of the two patients focusing on genes previously associated with a pro-thrombotic state, we did not find any mutation already described as associated with thrombophilia. Moreover, given the fact that β2GPI represents the main antigen to which aPL are directed, the same analysis has been carried out on *β2GPI* gene and its polymorphisms found in the two patients. In addition, in this case, patient 1 and patient 2 did not show any known or “new” mutations that could influence *β2GPI* function and thus that could have a role in anti-β2GPI antibodies development. All the observed *β2GPI* polymorphisms present in the patients were highly frequent in the general population and reported as “benign” variants.

The list of patient 1 and 2 genes of interest resulting from the whole exome sequencing and the screening process showed genes involved in the immune response and in the vascular homeostasis but that have never been associated with APS and, in general, that are not part of those known to be associated with a pro-thrombotic profile.

Overall, to some extent, these results can be seen as an additional proof of the complexity of this disease.

As aforementioned, the precise etiology of the syndrome is still unknown, but the high heterogeneity of APS manifestations and clinical course is presumably due to the occurrence of different mechanisms and alterations at different levels and pathways. This hypothesis would explain not only the complexity of APS but also the challenges faced until now in the study of its genetic background.

Given all these considerations, the network-based approaches seem worth exploring. The rationale of these approaches relies on the high interconnectivity characterizing the biological processes, which can be inter- or intra-cellular. The impact of a specific genetic alteration is not restricted to the activity of the gene product carrying it, but it can also alter products of genes that actually do not carry defects [83,84,85]. Thus, in order to better understand complex and multifactorial disorders, such as APS, and the consequences of genetic abnormalities, it is important to look at a gene as a part of a complex network of processes and interactions and not as an isolated entity.

Efforts to interpret the genetic risk factors involved in the heterogeneous clinical features of APS, for instance, the integration of the above-mentioned WES and network-based approaches, might help to identify and stratify patients at risk of developing APS. Our experience represents a pilot proof of concept, and international collaborations are warranted to confirm our observation.

Figure 2 summirizes the study work-flow.

## Figures and Tables

**Figure 1 ijms-21-09551-f001:**
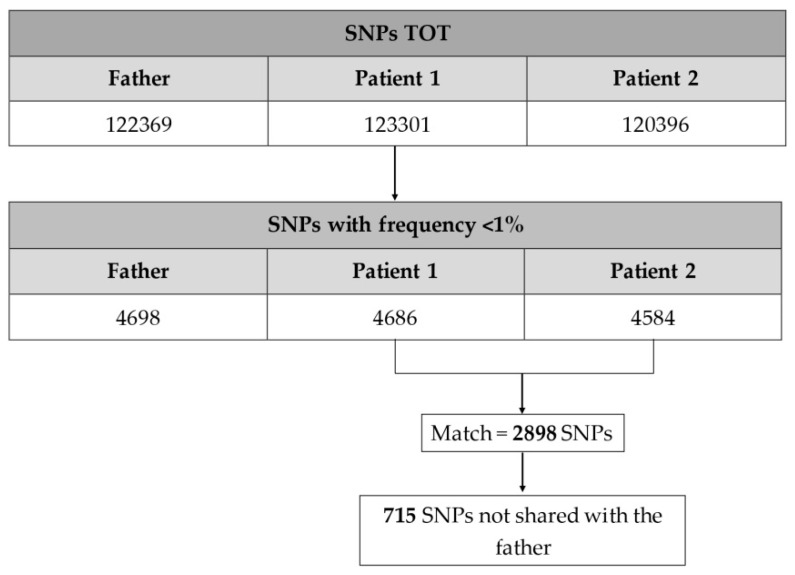
Schematic summary of the process used to filter the single-nucleotide polymorphisms (SNPs) obtained from the whole exome sequencing (WES) report.

**Figure 2 ijms-21-09551-f002:**
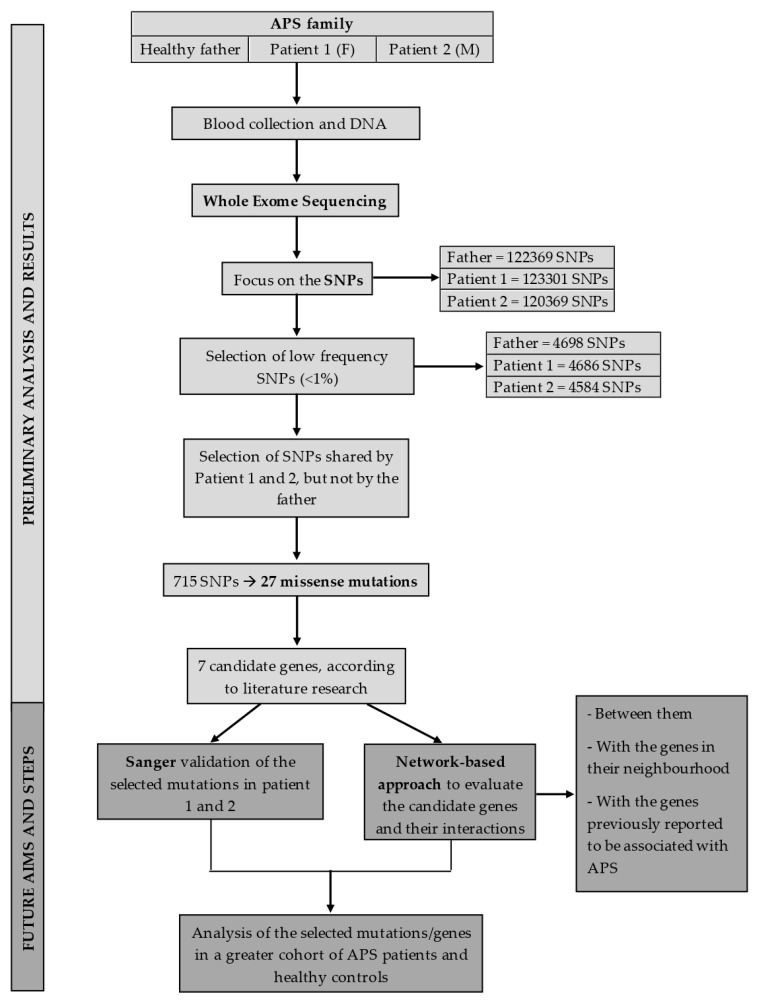
Summarizes the study work-flow and milestone. After the first screenings based on the variants frequency and the literature research of the genes carrying them, the presence of the selected single-nucleotide polymorphisms (SNPs) in patients 1 and 2 will be confirmed using Sanger sequencing. Then, these variants will be investigated in a validation cohort, consisting of 50 antiphospholipid syndrome (APS) patients [25 primary APS (PAPS) and 25 secondary APS (SAPS) patients] and a number of age/sex matched healthy controls in order to assess their frequency in APS. Given the multifactorial nature of this syndrome and the complex interplay within the biological systems, we also performed also a network analysis, using protein–protein interaction tools, to study the interactions between the proteins coded by the genes of interest, between them and their neighborhood, and between them and those coded by genes previously reported to be associated with APS in order to obtain a comprehensive perspective on APS development.

**Table 1 ijms-21-09551-t001:** Common mutations that have been associated with Antiphospholipid Syndrome.

Gene	Full Name	SNP	Chr	Coordinate	Transcript	Variant Effect	Ref
*BLK*	BLK proto-oncogene, Src family tyrosine kinase	rs2736340	8	11343973	NM_001715.3	5′ UTR variant	[34]
*B2GP1/APOH*	Beta-2-glycoprotein I, Apolipoprotein H	Position 247	17	64210757	NM_000042.3	Missense variant (p.Val247Leu)	[27,28]
*F2*	Coagulation factor II, Thrombin	rs1799963	11	46761055	NM_000506.5	3′ UTR variant	[50]
*F5*	Coagulation factor V	rs6025	1	169519049	NM_000130.5	Missense variant (p.Gln534Arg)	[50,51]
*GPIa/ITGA2*	Integrin subunit alpha 2	rs1126643	5	52347369	NM_002203.4	Synonymous variant (p.Phe253 =)	[40,41]
*GPIIIa/ITGB3*	Integrin subunit beta 3	rs5918	17	45360730	NM_000212.3	Missense variant (p.Leu59Pro)	[40,41]
*GP1BA*	Glycoprotein Ib platelet subunit alpha	rs2243093	17	4835895	NM_000173.7	5′ UTR variant	[41]
*IRF5*	Interferon regulatory factor 5	rs2070197	7	128589000	NM_001098629.3	3′ UTR variant	[34,35]
		rs10954213	7	128589427	NM_001098629.3	3′ UTR variant	[34]
*PROCR*	Protein C receptor	H1 haplotype	20		NM_006404.5		[42]
*PTPN22*	Protein tyrosine phosphatase non-receptor type 22	rs2476601	1	114377568	NM_015967.7	Missense variant (p.Arg620Trp)	[36,37]
*SELP*	Selectin P	rs6127	1	169566313	NM_003005.4	Missense variant (p.Asp603Asn)	[46]
*SERPINE1*	Serpin family E member 1	c.-817dupG	7	101126426	NM_000602.3		[47]
*STAT4*	Signal transducer and activator of transcription 4	rs7574865 rs3821236 rs3024866	2	191964633 191,902,758 191922841	NM_00124385.2 NM_00124385.2 NM_00124385.2	Intronic Intronic Intronic	[30,31,32,33,34]
*TFPI*	Tissue factor pathway inhibitor	intron 7 -33T>C	2	188385299	NM_006287.6	Intronic	[48]
*TLR4*	Toll-like receptor 4	rs4986790	9	12047502	NM_138554.5	Missense variant (p.Asp299Gly)	[38,39]
*TNF*	Tumor necrosis factor	rs361525	6	31543101	NM_000594.4	Upstream Transcript Variant	[49]

**Table 2 ijms-21-09551-t002:** Main clinical and laboratory characteristics of the patients included in the study.

Patient	Age	aPL Profile	Relevant Clinical History
1 (F)	51	Triple positive (LA, aCL IgG, aβ2GPI IgG)	Two episodes of ischemic stroke, one episode of CAPS (renal thrombotic microangiopathy, visual impairment, ischemic stroke)
2 (M)	47	Triple positive (LA, aCL IgG, aβ2GPI IgG)	Three episodes of deep vein thrombosis, regardless ongoing well conducted therapy vitamin k antagonist and additional retinal vein thrombosis

LA: lupus anticoagulant; aCL: anti-cardiolipin antibodies; aβ2GPI: anti-β2 glycoprotein I antibodies; CAPS: catastrophic APS.

**Table 3 ijms-21-09551-t003:** List of DNA missense variants of interest found in patient 1 and 2.

Gene	Full Name	SNP	Chr	Coordinate	Transcript	Variant Effect
***ADCY3***	**Adenylate cyclase 3**	**rs754839662 (C/T) Het**	**2**	**25050928**	**NM_004036.3**	**Missense variant (p.Val759Met)**
***ATP2B2***	**ATPase plasma membrane Ca^2+^ transporting 2**	**rs751257556 (G/A) Het**	**3**	**10382352**	**NM_001001331.2**	**Missense variant (p.Ala985Val)**
***BCL3***	**BCL3 transcription coactivator**	**rs747655476 (C/A) Het**	**19**	**45262063**	**NM_005178.4**	**Missense variant (p.Thr381Asn)**
*CC2D1A*	Coiled-coil and C2 domain containing 1A	c.794G>T Het	19	14028928	NM_017721.4	Missense variant (p.Arg265Leu)
*CROCC*	Ciliary rootlet coiled-coil/Rootletin	rs1444279934 (A/T) Het	1	17270657	NM_014675.3	Missense variant (p.Gln624Leu)
***CRTC3***	**CREB regulated transcription coactivator 3**	**c.163C>T Het**	**15**	**91083301**	**NM_022769.4**	**Missense variant (p.Leu55Phe)**
*FAT2*	FAT atypical cadherin 2	rs761199516 (C/A) Het	5	150945648	NM_001447.2	Missense variant (p.Ala949Ser)
*GRM7*	Glutamate metabotropic receptor 7	rs1480175679 (A/C) Het	3	7494339	NM_181874.2	Missense variant (p.Lys407Thr)
***HSPG2***	**Heparan sulfate proteoglycan 2**	**rs766963773 (C/T) Het**	**1**	**22207015**	**NM_005529.5**	**Missense variant (p.Arg679His)**
*IFNA17*	Interferon alpha 17	c.338A>T Het	9	21227835	NM_021268.2	Missense variant (p.Tyr113Phe)
*IMPA2*	Inositol monophosphatase 2	rs1423846345 (T/C) Het	18	12009980	NM_014214.2	Missense variant (p.Val110Ala)
*KIF14*	Kinesin family member 14	rs373895990 (C/T) Het	1	200573037	NM_014875.2	Missense variant (p.Arg598Gln)
*LTBP2*	Latent transforming growth factor beta binding protein 2	rs1310944162 (G/A) Het	14	74995323	NM_000428.2	Missense variant (p.Ala744Val)
*MAP3K10*	Mitogen-activated protein kinase kinase kinase 10	c.358G>A Het	19	40698296	NM_002446.3	Missense variant (p.Glu120Lys)
*MCM8*	Minichromosome maintenance 8 homologous recombination repair factor	rs768426546 (G/A) Het	20	5953350	NM_001281521.1	Missense variant (p.Arg451His)
*MRPL48*	Mitochondrial ribosomal protein L48	rs745995390 (G/T) Het	11	73555903	NM_016055.5	Missense variant (p.Asp85Tyr)
*MUC16*	Mucin 16	rs200972932 (C/T) Het	19	9067504	NM_024690.2	Missense variant (p.Glu6648Lys)
*OR2F1*	Olfactory receptor family 2 subfamily F member 1	rs777034277 (T/C) Het	7	143657370	NM_012369.2	Missense variant (p.Phe103Leu)
*OXNAD1*	Oxidoreductase NAD binding domain containing 1	rs1456594626 (T/G) Het rs1159857217 (T/A) Het rs1390159575 (C/G) Het	3	163431751634317616343177	NM_138381.3	Missense variant (p.Phe159Glu)
*PDZD2*	PDZ domain containing 2	rs1345334581 (C/G) Het	5	32098731	NM_178140.2	Missense variant (p.Pro2737Ala)
***PLA2G6***	**Phospholipase A2 group VI**	**rs780423461 (C/G) Het**	**22**	**38528920**	**NM_003560.2**	**Missense variant (p.Cys332Ser)**
*SRGAP3*	SLIT-ROBO Rho GTPase activating protein 3	rs764134718 (C/T) Het	3	9146464	NM_014850.3	Missense variant (p.Arg108Gln)
*USP32*	Ubiquitin specific peptidase 32	c.1169T>C Het	17	58313569	NM_032582.3	Missense variant (p.Leu390Pro)
***ZFAT***	**Zinc finger and AT-hook domain containing**	**rs748138009 (G/A) Het**	**8**	**135596174**	**NM_020863.3**	**Missense variant (p.Arg930Cys)**
*ZNF462*	Zinc finger protein 462	c.226A>T Het	9	109686419	NM_021224.4	Missense variant (p.Asn76Tyr)

Those with functions that may have a role in antiphospholipid syndrome pathogenesis are highlighted in bold.

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
