# Peer review of "Genetic Factors in Antiphospholipid Syndrome: Preliminary Experience with Whole Exome Sequencing"

_ijms, 2020, doi:10.3390/ijms21249551_

Round 1

Reviewer 1 Report

The article deals with the genetic factors associated with the appearance of APS. It is an interesting topic, it is an interesting subject, although the article has a strong limitation: only two cases are studied.

In the article there is an excellent introduction divided into two sections, a general introduction to APS and a comprehensive review of the “state of the art” about genetic factors in APS.

The third section consists of a joint description of methods and results. The descriptions are very interspersed making reading confusing. Reading would be easier and more systematic if it were divided into two sections: one for methods and one for results.

It is striking that after this third section the authors move on to a kind of final-section in which future perspectives are described. There is a lack of an extensive discussion section that in the manuscript is limited to lines in the macro section “methods and results” that make the reading even more confusing.

Since only two patients have been studied and there is no information on all the ancestors, nor on any descendants or indirect relatives, the information available on the genes studied in relation to the pathology of the patients is very limited. The lack of patients should be balanced by comparison with other studies in a powerful discussion section. The results obtained should be discussed extensively in relation to the results obtained from other studies on markers of genetic susceptibility to thrombosis, even if they are contradictory.

It is well known that the main antigen to which aPLs are directed is B2GP1.  As the authors have described in the introduction, some polymorphisms in the gene encoding B2GP1 may be related to the appearance of aPL. Therefore, it is striking that the authors do not describe the characteristics of the B2GP1 gene in these patients. This study has evidently been done since it has analyzed the complete exome that obviously includes B2GP1. It is mandatory that the authors include the analysis of the B2GP1 genotype of both patients and discuss whether the polymorphisms they present could be related to aPL. The presence of polymorphisms in genes that have been identified directly or indirectly in relation to autoimmune diseases, especially if it involves blood vessels, should also be discussed extensively. As an example, we highlight two of them:

1) perlecan (HSPG2) of which it has been described that the carboxy-terminal fraction (LG3) is a regulator of obliterative vascular remodeling and that Anti-LG3 autontibodies enhance microvascular damage in Ischemia-Reperfusion Injury especially in situations like kidney transplantation.

2) PLA2G6. Its role in relation to vascular pathology should be commented on since it has been described that autoantibodies against the PLA2 receptor bind in glomerular blood vessels (cell surface of podocytes) and participate in the pathogenesis of membranous glomerulonephritis.

Reviewer 2 Report

In this Review authors pointed to the importance of preliminary experience with whole exome sequencing in antiphospholipid syndrome with special consideration from environmental to genetic factors.

The Introduction section was well written with adequate citation of important parameters in the field of APS where etiology and genetic basis was additionally evaluated.

Genetics of APS was separately evaluated with special attention to familial and population studies. Authors elaborated up to dated findings.

Regarding APS family study section authors stated how the patient selection was done, as well as exome sequencing and variants filtering for tested patients. The genes that were included were separately analyzed and explained.

Authors explored future perspectives as well, with the possible addition to the existing knowledge of the results from familiar cases of APS.

Figures and tables are clear and understandable.

References are up to dated and representative.

Reviewer 3 Report

The authors provide a review of Antiphospholipid Syndrome and performed variant analysis based on exome sequencing. The manuscript has been partly improved through the revision; however, it is missing some details and has room for improvements. I do have some comments on the paper before I can recommend acceptance.

  1. Title; ‘From environmental to genetic factors in Antiphospholipid Syndrome’ There is no introduction or discussion about environmental factors. Please change the title according to the content.
  2. L135, L137; main mutations >>> common mutations
  3. Table 1 and 3. The genes would be shown in alphabetical 
  4. L141; A statement regarding the consent of the three donors to the publication of this study should be provided in the manuscript.
  5. L156; Please describe results for each sample from WES, which show mean depth and 10X coverage.
  6. WES data should be deposited to a genome database and the accession numbers would be provided.
  7. Table 3. Please provide zygosity of each mutation, described as homo or hetero.
  8. L338; the main antigen against which >>> the main antigen to which
  9. Abbreviations should be listed in alphabetical 
  10. Reference 17 is not properly described.

Round 2

Reviewer 1 Report

The questions and doubts raised have been satisfactorily resolved.
